# A Novel Efficient Convolutional Neural Algorithm for Multi-Category Aliasing Hardware Recognition

**DOI:** 10.3390/s22145358

**Published:** 2022-07-18

**Authors:** Yunzhi Zhang, Jiancheng Liang, Qinghua Lu, Lufeng Luo, Wenbo Zhu, Quan Wang, Junmeng Lin

**Affiliations:** School of Mechatronic Engineering and Automation, Foshan University, Foshan 528225, China; zhangyz@fosu.edu.cn (Y.Z.); 2112052027@stu.fosu.edu.cn (J.L.); luolufeng@fosu.edu.cn (L.L.); zhuwenbo@fosu.edu.cn (W.Z.); 2112052071@stu.fosu.edu.cn (Q.W.); 2112051004@stu.fosu.edu.cn (J.L.)

**Keywords:** convolutional neural networks, attention mechanisms, multi-category hardware, complex aliasing scenes

## Abstract

When performing robotic automatic sorting and assembly operations of multi-category hardware, there are some problems with the existing convolutional neural network visual recognition algorithms, such as large computing power consumption, low recognition efficiency, and a high rate of missed detection and false detection. A novel efficient convolutional neural algorithm for multi-category aliasing hardware recognition is proposed in this paper. On the basis of SSD, the novel algorithm uses Resnet-50 instead of VGG16 as the backbone feature extraction network, and it integrates ECA-Net and Improved Spatial Attention Block (ISAB): two attention mechanisms to improve the ability of learning and extract target features. Then, we pass the weighted features to extra feature layers to build an improved SSD algorithm. At last, in order to compare the performance difference between the novel algorithm and the existing algorithms, three kinds of hardware with different sizes are chosen to constitute an aliasing scene that can simulate an industrial site, and some comparative experiments have been completed finally. The experimental results show that the novel algorithm has an mAP of 98.20% and FPS of 78, which are better than Faster R-CNN, YOLOv4, YOLOXs, EfficientDet-D1, and original SSD in terms of comprehensive performance. The novel algorithm proposed in this paper can improve the efficiency of robotic sorting and assembly of multi-category hardware.

## 1. Introduction

In the past few decades, industrial robots have been extensively applied in hardware processing and manufacturing, which has improved the automation level of the industry [1]. However, the material sorting and feeding processes are still generally operated manually, because multi-category materials are stacked together in the processing workshop and the lighting conditions on site are always complicated, which makes the existing machine vision detection algorithms unable to work properly. When these algorithms are used to identify and locate hardware, the high misrecognition rate and poor positioning precision will restrict the application of industrial robots in the metal workpieces processing industry.

Thanks to the rapid development of computer image recognition technology in the past decades, convolutional neural network algorithms represented by Faster R-CNN [2], EfficientDet [3], YOLO [4,5], etc. have become the main research method in face recognition and target detection [6]. To improve the performance of visual inspection algorithms in robotic autonomous sorting and assembly operations, many researchers have carried out a series of studies. Brold et al. [7] proposed a method for synthesizing training data using spatial scanning and virtualized parts. This method improves the reliability of the data training process by combining deep learning and classical algorithms, and it realizes the accurate and rapid identification of auto parts based on small-sample datasets. Based on the YOLOv3 model, Chen et al. [8] improved the detection performance of non-ferrous metal targets based on small samples by optimizing the data enhancement method, improving the focal loss function and the IOU threshold function. The experimental results show that the recognition accuracy of the improved algorithm for aluminum scrap and copper scrap has reached 95.3% and 91.4%, respectively, and the algorithm runs at a speed of 18 FPS, which can meet the needs of real-time operation. Liu et al. [9] designed PolanNet-2d for 2D workpiece detection and PolishNet-3d deep neural network for 3D workpiece recognition, and they combined these two algorithms for polishing workpiece recognition methods. The experimental results show that the method performs well in the polishing workpiece recognition task. Yang et al. [10] designed a directional flipped image data augmentation algorithm and a multi-layer feature fusion network to improve YOLOv3 for the problems of the small amount of data and insufficient network feature extraction. The experimental results show that under the complex industrial background, the improved YOLOv3 network achieves strong robust real-time recognition of 0.8 cm thick needles and KR22 bearing machine parts. In order to solve the problems of difficult identification and classification of small-sample industrial mechanical parts, Li et al. [11] established a convolutional neural network model based on the InceptionNet-V3 pre-trained model through transfer learning. Through data expansion, adjusting the learning rate, and optimizing the algorithm, the optimal model was determined, which improved the recognition and classification performance of the algorithm for small-sample industrial machine parts, and the test accuracy rate reached 99.74%.

However, in the actual production site of robotic autonomous sorting and assembly, many types of spare parts with large differences in size are often piled up in a mess, and the on-site lighting conditions are usually complex and changeable, which seriously affects the recognition accuracy of existing visual inspection algorithms. In addition, although more network parameters can improve the detection accuracy of the algorithm, it will reduce the recognition efficiency of the algorithm and cannot meet the real-time detection requirements of industrial sites.

Therefore, in order to improve the recognition performance of existing visual detection algorithms, an efficient recognition algorithm for multi-category hardware in complex aliasing scenes is proposed in this paper. Compared with the previous algorithms, the main contributions and innovations of the new algorithm proposed in this paper are as follows:Proposed ISAB Spatial Attention Block, which is an improved spatial attention mechanism for SAM (spatial attention module in CBAM), in order to weight the effective features more efficiently for a very small increase in computational cost;Using ECA-Net to connect ISAB in series to form a new attention mechanism, where ECA-Net is a channel attention mechanism;Using the Resnet-50 instead of VGG16 as the basic feature extraction network to improve the performance of the backbone network, which not only can effectively extract deep features of objects but also speed up the convergence of the model and prevent exploding gradients when training the model;Experimental results show that the algorithm proposed in this paper can perform efficient (78 FPS) and accurate (98.20% mAP) real-time recognition of multi-category hardware in complex overlapping scenes.

## 2. Original SSD Network Structure

Single Shot MultiBox Detector (SSD) [12] is a representative single-stage detector, which consists of a VGG16-based backbone structure and extra feature layers. It combines the advantages of Faster R-CNN and YOLO, and it has a great improvement in model structure and operation speed. The network structure of the original SSD is shown in Figure 1.

As shown in Figure 1, the backbone of SSD consists of the first five layers of convolutional networks of VGG16, and extra feature layers contain five feature extraction layers (Conv6, Conv7, Conv8, Conv9, and Conv10). Therefore, the entire network contains six feature extraction layers finally. Each feature extraction layer needs to be passed to the detector for frame regression and classification, and then, the non-maximum suppression (NMS) algorithm is used to eliminate candidate recognition frames with low confidence to achieve target recognition. The algorithm principle of NMS is shown as Equation (Equation 1).
(1)$$s_{i}=\left\{\begin{array}{cc} s_{i}, & \mathrm{iou}\left(M,b_{i}\right)<N_{i}\\ 0, & \mathrm{iou}\left(M,b_{i}\right)\geq N_{i} \end{array}\right.$$
where $s_{i}$ represents the score of each recognition frame, *M* is the current recognition frame with the highest score, $b_{i}$ is one of the remaining recognition frames, and $N_{i}$ is the set threshold.

It can be seen from the formula that when the Intersection over Union (IoU) is greater than $N_{i}$, the score of the recognition frame is directly set to 0, which is equivalent to being discarded, so that the recognition frame with the highest confidence is selected.

For the COCO and Pascal VOC datasets, although SSD has a high level of operation speed and detection accuracy, compared with other networks (such as YOLOv4, YOLOX, and EfficientDet), the lack of convolutional layers will lead to the problem of insufficient target semantic feature extraction.

## 3. Proposing Novel Algorithm

The backbone network of the original SSD network (VGG16) has a deficiency of feature extraction capability due to the small number of convolutional layers and the general loss of feature information during the information transfer in the ensemble layer. In order to improve the performance of the original SSD algorithm for multi-category parts feature extraction, this paper proposes an improved SSD algorithm with the structure shown in Figure 2. The first part is the backbone, which is composed of the first four feature layers of Resnet-50. The second part is the novel attention mechanism proposed in this paper. The third part is the extra feature layer, where the features weighted by the attention mechanism are convoluted five times to obtain five extra layers of features. Finally, the output of the backbone and the extra feature layers (Conv6, Conv7, Conv8, Conv9, and Conv10) are sequentially transmitted into the detection network for regression and classification, and the NMS algorithm filters out the highest scoring detection targets.

### 3.1. Replacing the Backbone Network

The backbone is used to extract the initial target features, and the inputs of the subsequent feature extraction layers and the target detection network are based on the features extracted by the backbone. Therefore, the performance of the backbone network determines the performance of the whole algorithm. Compared with VGG16 [13], ResNet [14] utilizes the residual structure to make the network deeper, faster to converge, easier to optimize, and less complex. It can effectively extract target features and better deal with network degradation. Inspired by [15,16], we used the first four convolutional layer structures of Resnet-50 (Conv1, Conv2, Conv3, Conv4) and adjusted the stride of Block1 of Conv4 (in order to keep the output feature size consistent with the original SSD), thus forming a Resnet-50-based backbone network with the structure shown in Figure 3, to replace VGG16 as the backbone network of the algorithm proposed in this paper.

### 3.2. Integrating Attention Mechanism

The attention mechanism proposed in this paper is composed of two attention modules, ECA-Net [17] and ISAB (Improved Spatial Attention Block). As shown in Figure 4, the ECA-Net is a channel attention mechanism and ISAB is a spatial attention mechanism. Referring to the connection between the channel attention mechanism and the spatial attention mechanism of CBAM [18], the accuracy of target recognition can be improved. The attention mechanism proposed in this paper adopts the serial connection mode of ECA-Net before ISAB and after.

#### 3.2.1. ECA-Net

ECA-Net is an improved SE-Net [19] attention mechanism network. Compared with SE-Net, ECA-Net removes the fully connected layer in SE-Net and directly uses one-dimensional sparse convolution operation on the features after global average pooling to optimize the fully connected layer operation while maintaining network performance. At the same time, the number of parameters of ECA-Net is greatly reduced.
(2)$$k=\psi \left(C\right)=\frac{\log_{2}C+1}{2}$$
(3)$$\mathrm{Sigmoid}\left(x\right)=\frac{1}{e^{-x}+1}$$

The implementation idea of ECA-Net is divided into the following three steps, as shown in Figure 5:The original feature map with the input size of H×W×C generates feature maps with the size of 1×1×C through global average pooling;Calculate the adaptive convolution kernel size with Equation (Equation 2);Calculate the activation value of the one-dimensional convolution output with sigmoid Equation (Equation 3) and obtain the weight of each channel.

#### 3.2.2. Improved Spatial Attention Block (ISAB)

Inspired by CBAM [18], this paper improves the SAM (Spatial Attention Module) of CBAM to Improved Spatial Attention Block (ISAB), whose architecture is shown in Figure 6, and the comparison between SAM (Spatial Attention Module in CBAM) and ISAB is shown in Table 1. Compared with SAM, ISAB also performed Max Pooling and Agv Pooling on the feature map $\tilde{\chi }=\left(H,W,C\right)$ output by ECA-Net, and it obtained ${\tilde{\chi }}_{1}$ and ${\tilde{\chi }}_{2}$ with the dimensions (H,W,1) of two feature maps. In order to obtain rich spatial features, ${\tilde{\chi }}_{1}$ and ${\tilde{\chi }}_{2}$ are transposed to add two feature maps of size (W,H,1) and obtain $\left({\tilde{\chi }}_{1},{\tilde{\chi }}_{2},{\tilde{\chi }}_{1}^{T},{\tilde{\chi }}_{2}^{T}\right)$ four feature maps. We concatenate these feature maps into a feature map group $F_{C}$ and then use the sigmoid function to obtain the weight of each feature point of the input feature layer. Finally, we use the feature weight $F_{\sigma }$ to multiply the original input feature maps $\tilde{\chi }$ to obtain the weighted feature of ISAB.

## 4. Design Experiment and Performance Metrics

### 4.1. Building the Dataset

In order to simulate the robotic autonomous sorting and assembly scene of multi-category metal parts, three types of hardware with different sizes are chosen and aliased together, and then, a dataset of multi-category hardware aliasing has been constructed by some data augmentation methods. These three hardware (A, B, and C) included in this dataset are shown in Figure 7a, simulating a cluttered stack of hardware datasets for an industrial site, as shown in Figure 7b.

To construct the original dataset, 300 images are captured with a resolution of 1280×720, and the shooting distance is set to 0.6–1.2 m, while the light source is the ordinary LED white light. Each image contains 15–35 shielding metal parts. Subsequently, referring to the paper [20], some dataset augmentation methods (including rotation, adjusted hue, changed saturation, and brightness, as shown in Figure 8) are used to expand the dataset to 1500 and to improve the generalization ability of the trained network model finally.

### 4.2. Hyperparameter and Performance Metrics Settings

The experiment uses a Windows10 system, CUDA11.1, CUDNN8.0.5, Pytorch1.8 deep learning framework. The hardware configuration is an Intel(R) Core i5-11400 CPU@2.60 GHz processor, 16 G memory, and the GPU is NVIDIA GeForce RTX 3060TI. The model training parameters are listed in Table 2.

Referring to the paper [21,22], by analyzing and comparing the framework structures and operating mechanisms of existing visual detection algorithms, and considering both the hardware costs and the deployment of models used in real industrial scenarios, the AP, mAP, FPS, FLOPs, and GPU memory footprint are chosen as the performance evaluation metrics of deep neural network algorithms in this paper. The specific introduction and calculation formula of each metric are shown as follows:Precision is the identified sample, the ratio of true positives, as shown in Equation (Equation 4);
(4)$$\mathrm{Precision}=\frac{\mathrm{TP}}{\mathrm{TP}\;+\;\mathrm{FP}}$$

Recall is the percentage that is correctly recognized as true positives in the test set, as shown in Equation (Equation 5);
(5)$$\mathrm{Recall}=\frac{\mathrm{TP}}{\mathrm{TP}\;+\;\mathrm{FN}}$$
where TP, FP, and FN are true positives, false positives, and false negatives, respectively.

AP is the average for each category of precision, as shown in Equation (Equation 6);
(6)$$\begin{array}{c} \mathrm{AP}=\sum_{i=1}^{n-1}\left(r_{i+1}-r_{i}\right)P_{\mathrm{inter}\;}\left(r_{i}+1\right)\\ P_{\mathrm{inter}\;}\left(r_{i}+1\right)=\max_{\tilde{r}:\tilde{r}\geq {\tilde{r}}_{i+1}}\left(P\left(\tilde{r}\right)\right) \end{array}$$
where $P(\tilde{r})$ is the measured precision at recall $\tilde{r}$.

The mAP refers to the mean of all categories of AP, as shown in Equation (Equation 7).


(7)
$$\mathrm{mAP}=\frac{\sum_{i=1}^{k}AP_{i}}{k}$$


FPS is the number of images processed by the algorithm per second (the number of images completed for recognition), as shown in Equation (Equation 8); the larger the value, the faster the algorithm processes.
(8)$$\mathrm{FPS}=\frac{1}{T}$$
where T is the time taken to process a frame.

In practice, FPS focuses more on describing the adaptability between the algorithm and the deployed devices, which means that the FPS value of the same algorithm deployed on different devices may be very different.

FLOPs (Floating Point Operations) are the sum of the total computation of the entire model, which depends on, but not entirely on, the complexity of algorithm structure.GPU memory footprint is the amount of memory used by the dedicated GPU for model operations.

## 5. Experimental Results and Analysis

### 5.1. Heatmaps Analysis

In order to visualize the capability of the algorithm in terms of target feature information extraction with different backbone networks and the addition of attention mechanisms, this paper uses Grad-CAM [23] to generate algorithm learning heatmaps to visually represent the target feature regions extracted by the network. The specific results are shown in Figure 9.

From the figure, it can be visualized that the original SSD algorithm (a) extracts the target feature information region showing a scattered state and focuses on useless features (background and image edges), leading to ignoring much of the target feature information. After replacing VGG-16 with Resnet-50 as the backbone network (b), the ability to extract features is significantly improved, but there are still a large number of redundant features. After that, the features extracted from the backbone network are weighted and filtered by using the existing attention mechanisms ECA-Net as well as SAM in the form of ECA-Net and ECA-Net+SAM, respectively. The heatmap (c) shows that the addition of ECA-Net can eliminate a large number of invalid features and improve the efficiency of feature extraction, but there are some parts with too few learning regions. The heatmap (d) shows that adding ECA-Net+SAM can involve more features and give higher weights than adding only ECA-Net, but there is a misconception of giving higher weights to the background. Therefore, the attention mechanism of ECA-Net+ISAB is proposed in this paper. It can be seen from the heatmap (e) that the region learned by the model after adding ECA-Net+ISAB is gradually concentrated in the part region and the redundant feature information is gradually reduced, indicating that the neural network proposed in this paper can better concentrate on the target learning region and reduce the interference of redundant feature information. The final experiments demonstrated that Resnet-50 and the attention mechanism are helpful to improve the learning ability of the SSD algorithm.

### 5.2. Comparison of Recognition Performance

Subsequently, this paper compares the recognition performance of several existing representative deep neural network algorithms (Faster R-CNN, YOLOv4, YOLOXs, EfficientDet-D1, SSD) and improved deep neural network algorithms based on SSD for a typical multi-category metal parts aliasing scene. The experimental results are divided into the recognition accuracy performance (Table 3) of the algorithm for each part, the overall recognition performance of the algorithm (Table 4), and the actual recognition effectiveness of the algorithm (Figure 10).

Since the actual industrial scenario is oriented to parts with multi-category and significant dimensional differences, the balance of AP is extremely important. Comparing the AP in Table 3, it is easy to see that the new algorithm proposed in this paper improves the detection precision of A, B, and C parts compared to other SSD-based algorithms, especially for small target class C. The AP of the proposed algorithm for multi-target parts detection is already comparable to the YOLO series algorithms and exceeds that of the classical deep neural network algorithm, Faster R-CNN. This improvement is obvious and shows the effectiveness of the algorithm proposed in this paper.

The actual deployment of the algorithm in the application considers not only the accuracy but also the efficiency of the algorithm operation and the cost of the algorithm deployment in the device. It can be seen from Table 4 that replacing the backbone of SSD from VGG-16 to Resnet-50 can significantly reduce the FLOPs and the GPU memory footprint of the algorithm, the FLOPs are reduced from 30.59 to about 15.12, and the GPU memory footprint is reduced from 3.6 to about 2.3 GB (all the value of GPU memory footprints are measured at batch size = 4). Compared with other improved algorithms based on SSD, the FLOPs of the algorithm proposed in this paper are increased by only an extremely limited amount, which is mainly caused by the convolutional operations (such as these Conv layers shown in Figure 6) in the attention mechanism. In addition, in the attention mechanism, when the feature is weighted by the sigmoid function, it will be copied once in tensor format, and this operation will increase the GPU memory footprint of an algorithm. In other words, each time the attention mechanism is added, the GPU memory footprint of an algorithm will increase a little. Therefore, the algorithm proposed in this paper and SSD+ResNet-50+ECA+SAM have the most amount of GPU memory footprint, SSD+ResNet-50+ECA comes second, and SSD+ResNet-50 has the least. So, according to the FLOPs and GPU memory footprint, we can conclude that the algorithm proposed in this paper can improve the performance with almost no increase in complexity.

Figure 10 shows the recognition results of the nine algorithms. Comparing the detection results in the figure, the detection algorithm proposed in this paper not only has higher recognition accuracy and confidence for parts in complex and cluttered scenes but also locates the parts more accurately and has better detection performance for small objects. From (i), it can be seen that the ECA-Net+ISAB attention mechanism has a greater improvement than ECA-Net in the overall recognition performance of the network. The experimental results show that the algorithm proposed in this paper has superior overall performance to the other algorithms.

## 6. Conclusions

In order to improve the multi-category aliasing targets recognition performance of existing visual detection algorithms in the scene of industrial robotic autonomous sorting and assembly, a novel algorithm, based on the original SSD network, is proposed in this paper. First, it replaces VGG-16 in the original network with ResNet-50 with deeper network layers and faster convergence. Subsequently, attention mechanisms such as ECA-Net and ISAB are integrated to optimize the base SSD network base on ResNet-50 and improve the multi-target recognition performance of the algorithm. By constructing an augmented dataset and comparing the detection performance of Faster R-CNN, YOLOv4, YOLOXs, EfficientDet-D1, original SSD, and improved SSD networks for metal parts in complex cluttered scenes, the model proposed in this paper significantly improves the detection performance of small-sized objects (AP increased from 88.66 to 97.88), improves the overall detection performance (mAP is 98.20%), and maintains a high detection speed (the FPS is 78).

The experimental results show that the multi-category target detection performance of the algorithm proposed in this paper is close to YOLOv4 and EfficientDet-D1, but the detection efficiency is much higher than both, which can better meet the requirements of real-time operation in industrial sites. The research work in this paper can provide a theoretical basis and reference for improving the autonomous sorting and assembly capabilities of industrial robots for multiple types of metal parts.

Although the model proposed in this paper performs well in simulated industrial scenarios, it has the following shortcomings or limitations. The existing problems will continue to be researched in subsequent studies:The training model requires a large amount of data, and the labeling of the dataset is labor-intensive;The attention mechanism proposed in this paper can eliminate redundant features, but it also ignores some features of the target object;Although the model proposed in this paper runs faster, it still needs to be lightly embedded in industrial equipment;The dataset for validating the algorithm consists only of metal parts, and there may be limitations for the performance of non-metal parts;The attention mechanism proposed in this paper, ECA-Net+ISAB, is extremely dependent on the feature capability of the backbone network, which may have limitations for the optimization effect of lightweight and complex backbone networks.

## Figures and Tables

**Figure 1 sensors-22-05358-f001:**
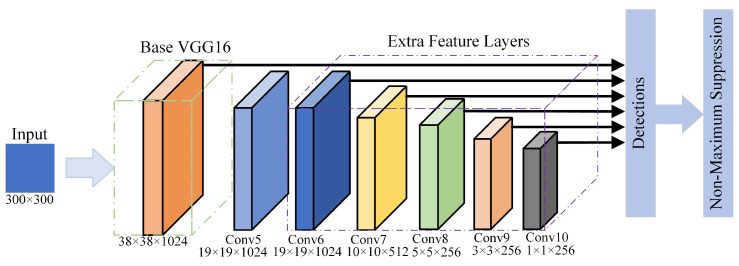
Structure of SSD algorithm.

**Figure 2 sensors-22-05358-f002:**
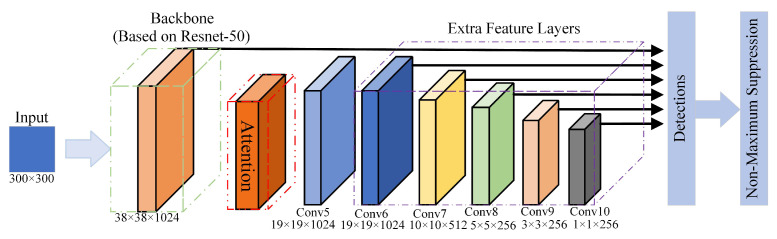
Improved SSD algorithm architecture.

**Figure 3 sensors-22-05358-f003:**
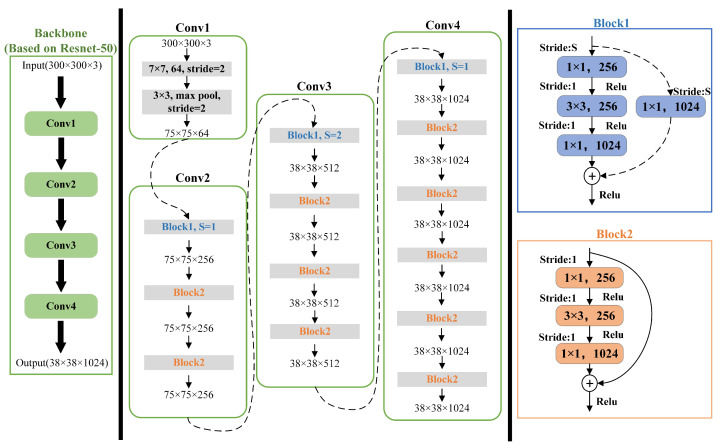
The structure of the backbone (based on Resnet-50) proposed in this paper.

**Figure 4 sensors-22-05358-f004:**
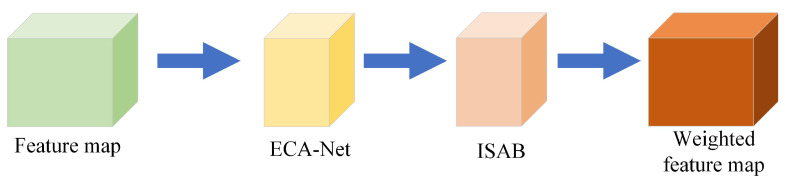
The structure of the attention mechanism proposed in this paper.

**Figure 5 sensors-22-05358-f005:**
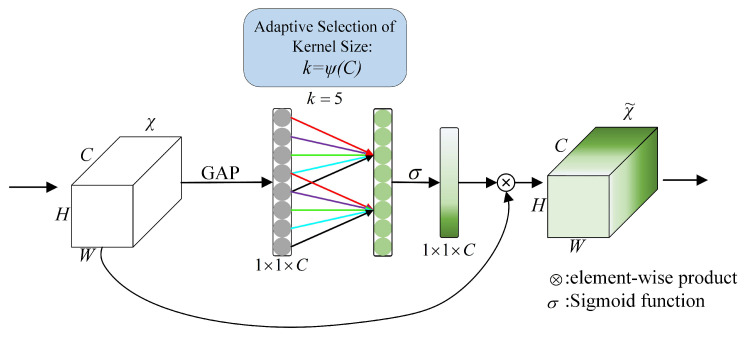
Structure of ECA-Net.

**Figure 6 sensors-22-05358-f006:**
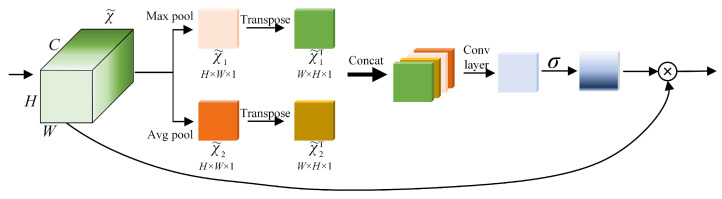
Structure of Improved Spatial Attention Block.

**Figure 7 sensors-22-05358-f007:**
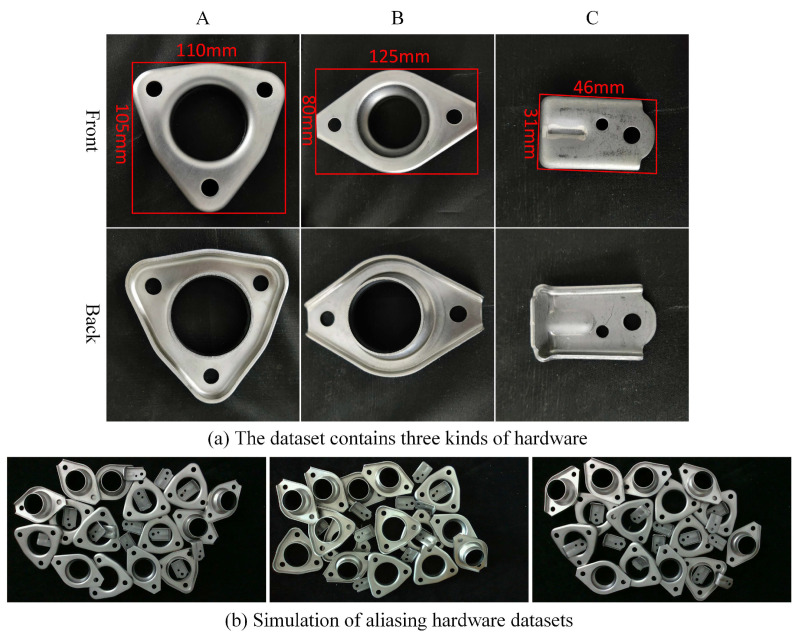
Dataset augmentation.

**Figure 8 sensors-22-05358-f008:**
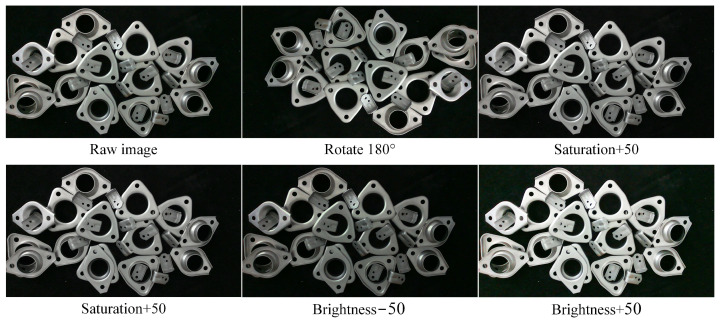
The details of hardware included in the dataset.

**Figure 9 sensors-22-05358-f009:**
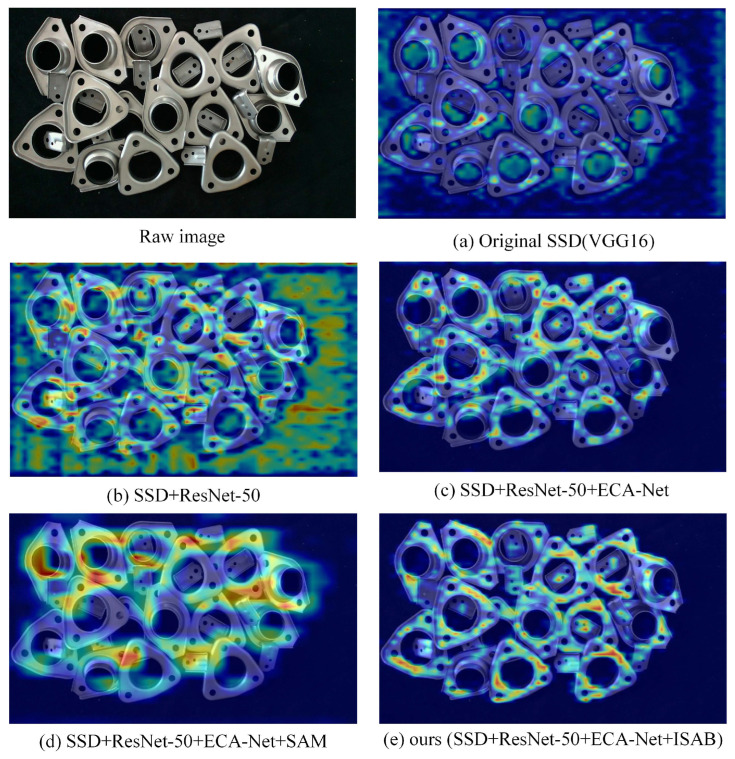
Heatmap of network learning area before and after improvement. (The brighter areas in the graph represent the higher weights assigned by the algorithm.)

**Figure 10 sensors-22-05358-f010:**
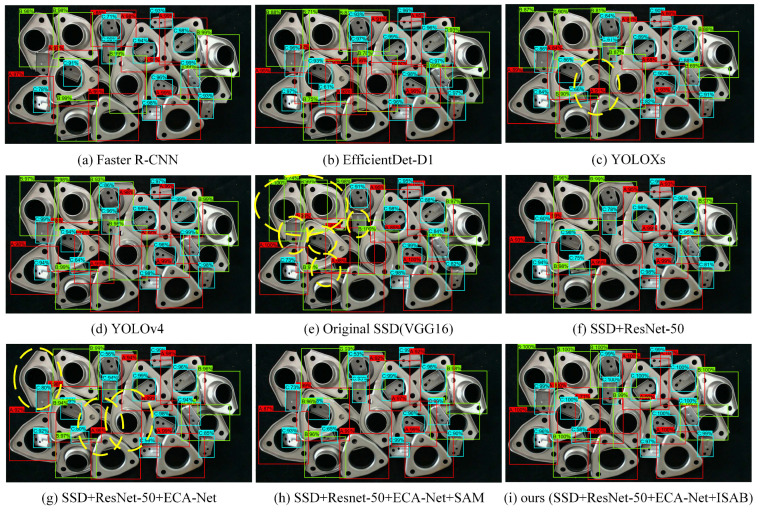
The actual results of six algorithms for metal parts recognition are shown. (The yellow ellipse dotted line in the figure is the model’s misidentified or missed identification.)

**Table 1 sensors-22-05358-t001:** Comparison of SAM and ISAB.

SAM	ISAB
Input $\tilde{\chi }=\left(H,W,C\right)$	Input $\tilde{\chi }=\left(H,W,C\right)$
Max Pooling $\left(\tilde{\chi }\right)={\tilde{\chi }}_{1}$	Max Pooling $\left(\tilde{\chi }\right)={\tilde{\chi }}_{1}$
Agv Pooling $\left(\tilde{\chi }\right)={\tilde{\chi }}_{2}$	Agv Pooling $\left(\tilde{\chi }\right)={\tilde{\chi }}_{2}$
Concat $\left({\tilde{\chi }}_{1},{\tilde{\chi }}_{2}\right)=F_{C}$	Transpose $\left({\tilde{\chi }}_{1},{\tilde{\chi }}_{2}\right)=\left({\tilde{\chi }}_{1}^{T},{\tilde{\chi }}_{2}^{T}\right)$ and Concat $\left({\tilde{\chi }}_{1},{\tilde{\chi }}_{2},{\tilde{\chi }}_{1}^{T},{\tilde{\chi }}_{2}^{T}\right)=F_{C}$
Sigmoid (Conv$\left(F_{C}\right))=F_{\sigma }$	Sigmoid (Conv$\left(F_{C}\right))=F_{\sigma }$
Output $F_{\sigma }\times \tilde{\chi }$	Output $F_{\sigma }\times \tilde{\chi }$

**Table 2 sensors-22-05358-t002:** Model training hyperparameter.

Hyperparameter	Value
Input size	300×300
Learning rate	0.0005
Weight decay	0.0005
Batch size	4
Epochs	200
Momentum	0.9
Gamma	0.9
Optimizer	Adam

**Table 3 sensors-22-05358-t003:** Average precision (AP) comparison of different models.

Methods	A/%	B/%	C/%
Faster R-CNN	98.34	98.54	90.79
YOLOv4	98.31	98.27	97.87
YOLOXs	98.00	97.18	98.07
EfficientDet-D1	98.12	97.41	99.55
Original SSD	98.24	97.95	88.66
SSD+ResNet-50	98.52	97.87	94.31
SSD+ResNet-50+ECA	98.43	96.63	96.43
SSD+ResNet-50+ECA+SAM	98.40	97.96	97.82
Ours	98.50	98.22	97.88

**Table 4 sensors-22-05358-t004:** Performance comparison of different models. (FLOPs and GPU memory footprint are measured by torchstat-0.0.7 and NVIDIA-SMI-472.12, respectively.)

Methods	mAP/%	FPS	FLOPs /GFLOPs	GPU Memory Footprint /GB
Faster R-CNN	95.89	18	184.99	6.6
YOLOv4	98.15	42	29.89	6.3
YOLOXs	97.75	62	13.32	2.0
EfficientDet-D1	98.36	20	11.21	3.2
Original SSD	94.95	91	30.59	3.6
SSD+ResNet-50	96.90	89	15.12	2.3
SSD+ResNet-50+ECA	97.83	81	15.12 + 0	2.4
SSD+ResNet-50+ECA+SAM	98.06	80	15.12 + 0.00025	2.5
Ours	98.20	78	15.12 + 0.00051	2.5

## Data Availability

The data are not publicly available due to the project requirements.

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
