# Peer review of "A Novel Efficient Convolutional Neural Algorithm for Multi-Category Aliasing Hardware Recognition"

_sensors, 2022, doi:10.3390/s22145358_

Round 1

Reviewer 1 Report

The submission describes a Convolutional Neural Algorithm for Multi-category Aliasing Hardware Recognition. A backbone SSD Network is extended and improved. The presented results show the proposed method's improved performance compared to comparable approaches. The overall quality of the paper is in line with the expectation. Following are some of the comments which should be addressed in the revised version of this paper:

1.   The contribution and innovation of this approach over previous schemes should be clearly stated at the end of the first section.

2.   The proposed neural network is not adequately described. I suggest including an algorithmic representation.

3.    In experiments, I suggest that the authors add more results than the other comparable approaches introduced. Besides, the explanation of experimental results is not enough.

4.   If there are limitations in the proposed approach, I recommend discussing the limits of the approach in the Discussion section.

Reviewer 2 Report

The paper tackles the problem of workpiece recognition. The scenario addressed by the authors may be relevant in robotic automation for manufacturing: the goal is to exploit computer vision to detect different types of workpieces that are mixed together.

The proposed pipeline involves standard models. The only exception is ISAB, a block that implements spatial attention; in fact, ISAB augments a state-of-the-art model (SAM). Nonetheless, the experimental results are interesting.

Major remarks:

11)      Experimental results: the comparison should include the pipeline SSD+ResNet+ECA+SAM. Since the authors are proposing to replace SAM with ISAB, the question is: does SSD+ResNet+ECA+ISAB perform better than SSD+ResNet+ECA+SAM? The paper should reply to this question.

22)      The authors also pose the problem of real-time detection. Accordingly, they replace VGG16 with ResNet-50 in their pipeline. I think that the paper should compare the different solutions in terms of FLOPS and memory footprint. Such analysis would help the reader to better characterize the proposed pipeline. I would also suggest the authors to take into consideration the MobileNet architecture, which is a lightweight model that proved able to achieve outstanding performance on object detection.

Round 2

Reviewer 2 Report

The authors have improved the overall quality of the paper. However, I have concerns about the quantities FPS, FLOPS, and memory footprint (i.e., memory usage). 

FPS is a quantity that depends on the hardware on which the model is deployed. That is, the same model can feature different values of FPS when deployed on different computer systems (e.g., different GPUs). Conversely, FLOPS and memory usage are quantities that characterize the model itself. Thus, these quantities are not at the same level. In this regard, I think that Table 4 should be carefully checked. For example, in the case of Faster R-CNN the value "6 GFLOPS" seems not correct.

The authors should also explain how FLOPS and memory usage have been assessed for SSD+ResNet-50+ECA, SSD+ResNet-50+ECA+SAM, and SSD+ResNet-50+ECA+SAM+ISB. 

Round 3

Reviewer 2 Report

Before submitting the final version of the manuscript the authors should:

1) add information about the tool adopted to assess GFLOPS and memory footprint;

2) thoroughly check English language.
